# Diaphragm thickness and mobility elicited by two different modalities of inspiratory muscle loading in heart failure participants: A randomized crossover study

Tatiana Zacarias Rondinel [1]*, Lilian Bocchi[2], Gerson Cipriano Júnior[2‡], Gaspar Rogério da Silva Chiappa[3‡], Gabriela de Sousa Martins[2], Sérgio Ricardo Menezes Mateus [4], Lawrence Patrick Cahalin[5], Graziella França Bernardelli Cipriano[2]

1 Science and Technology in Health Program, University of Brasília, Brasília, Distrito Federal, Brazil, 2 Physical Therapy Department, Science of Rehabilitation Program, University of Brasília, Brasília, Distrito Federal, Brazil, 3 Centro Universitário de Anápolis, Anápolis, Goiás, Brazil, 4 Physical Therapy Department, University of Brasília, Brasília, Distrito Federal, Brazil, 5 Department of Physical Therapy, Leonard M. Miller School of Medicine, University of Miami, Coral Gables, Florida, United States of America

☯ These authors contributed equally to this work.
‡ These authors also contributed equally to this work
* tattyrondinel@hotmail.com

## Abstract

### Objectives

To analyze diaphragmatic thickness, at end-inspiration and end-expiration, diaphragmatic thickening index and mobility via US under two different modalities of inspiratory muscle loading, in two different modalities of inspiratory muscle loading and different load intensities at full-vital capacity maneuvers and the relationship between diaphragmatic thickness with pulmonary function tests in participants with HF.

### Methods

This randomized crossover trial, enrolled with 17 HF subjects, evaluated diaphragm thickness (Tdi, mm), fractional thickness (TFdi, %), and mobility (mm) US during low and high intensities (30% and 60% of maximal inspiratory pressure—MIP) with two modalities of inspiratory muscle loading mechanical threshold loading (MTL) and tapered flow-resistive loading (TFRL).

### Results

Both MTL and TFRL produced a increase in Tdi, but only with high intensity loading compared to baseline—2.21 (0.26) *vs.* 2.68 (0.33) and 2.73 (0.44) mm; p = .01. TFdi was greater than baseline under all conditions, except during low intensity of TFRL. Diaphragm mobility was greater than baseline under all conditions, and high intensity of TFRL elicited greater mobility compared to all other conditions. Additionally, baseline Tdi was moderately correlated with pulmonary function tests.

**Data Availability Statement:** "All relevant data are within the paper and its Supporting information files".

**Funding:** Funding Agency: Fundação de Amparo à Pesquisa do Distrito Federal (FAPDF) Grant Number: 00193-00000861/2023-36 awarded to G. F.B.C.

**Competing interests:** The authors have declared that no competing interests exist.

## Conclusions

MTL and TFRL modalities elicit similar increases in diaphragm thickness at loads, but only during high intensity loading it was greater than baseline. Diaphragm mobility was significantly greater than baseline under both loads and devices, and at high intensity compared to low intensity, although TFRL produced greater mobility compared to modalities of inspiratory muscle loading. There is an association between diaphragm thickness and pulmonary function tests.

## Introduction

Patients with heart failure (HF) often develop skeletal myopathies, which may affect both limb and respiratory muscles [1]. Respiratory muscle decline can lead to several common manifestations in HF, including dyspnea, fatigue, exercise intolerance, and poor functional status and quality of life [1–3].

The diaphragm plays a major role supporting ventilatory capacity in different situations and tasks, including exercise. Therefore, diaphragmatic evaluation is considered extremely relevant, particularly for patients with chronic diseases [4,5]. Different methods to assess diaphragm function have been used, presenting differing levels of clinical utility and diagnostic accuracy [6,7], with specifics advantages and disadvantages [8].

Recent studies have recognized the clinical utility of ultrasonography (US) as a method to evaluate diaphragm function in several patient populations [4,6,9]. In HF, only one study demonstrated that diaphragm dysfunction (thickness at the end of inspiration <4mm) was associated with exercise intolerance (lower gate speed and fewer meters covered in the 6-minute walk test) and inspiratory muscle weakness (lower maximal inspiratory pressure values) [6].

As a cost-effective non-pharmacological treatment, inspiratory muscle training (IMT) provides benefits in HF patients, such as improvement of limb blood flow, and an increase in exercise capacity, peak oxygen consumption, inspiratory muscle strength, and quality of life [10–12]. The most common IMT method is mechanical threshold loading (MTL), which uses a calibrated spring to provide a threshold load of resistance during inspiration [10]. The MTL action occurs when the user generates a pressure that is greater than that at which the tension of the spring is set, in other words the valve opens when a resistance occurs against this loaded spring, during the inspiratory maneuver [13,14].

Recently, a new modality of inspiratory muscle loading has emerged as an electronic tapered flow-resistive loading (TFRL) device [14,15]. Tapered flow-resistive loading incorporates a valve that dynamically adjusts the resistance, to maintain the pressure load across the entire inspiratory maneuver [16]. However, a better understanding of loading [3] and different muscle activation, recruitment, and training of different types of muscles is required to optimize IMT performed using MTL and TFRL.

Therefore, the aims of this study were to examine the diaphragmatic thickness, diaphragm thickening fraction and mobility, using US, according to two different modalities of inspiratory muscle loading (MTL and TFRL); and to investigate the relationship between diaphragmatic thickness and the pulmonary function and respiratory muscle function test in participants with HF using two different inspiratory loadings (low and high), at full-vital capacity inspiration starting from residual volume.

We confirm the hypotheses that these different modalities of inspiratory muscle loading at different load intensities would generate changes in diaphragm thickness and mobility as a

result of the inspiratory muscle loading and intensity of loading, and that these would be related to lung function and the respiratory muscle strength test.

## Material and methods

This prospective, randomized, crossover study, included 17 HF participants recruited from the Cardiac Rehabilitation Program at the University of Brazilia, from January 5 to June 30, 2018. The study protocol was approved by the ethics committee of University of Brazilia (CAAE: 67204717.7.0000.8093). This study was registry at REBEC (ensaiosclinicos.gov.br)–RBR-5sfmz7. All participants signed a written consent form. The study is summarized in the CONSORT flow diagram (Fig 1).

Inclusion criteria were clinical HF diagnosis [17,18], and optimized treatment of HF at least three months prior to entering the study. Exclusion criteria were angina or myocardial infarction; neurologic, orthopedic, or infectious diseases; and participants receiving corticosteroid or hormones treatments, or cancer chemotherapy. Participants with pulmonary limitations

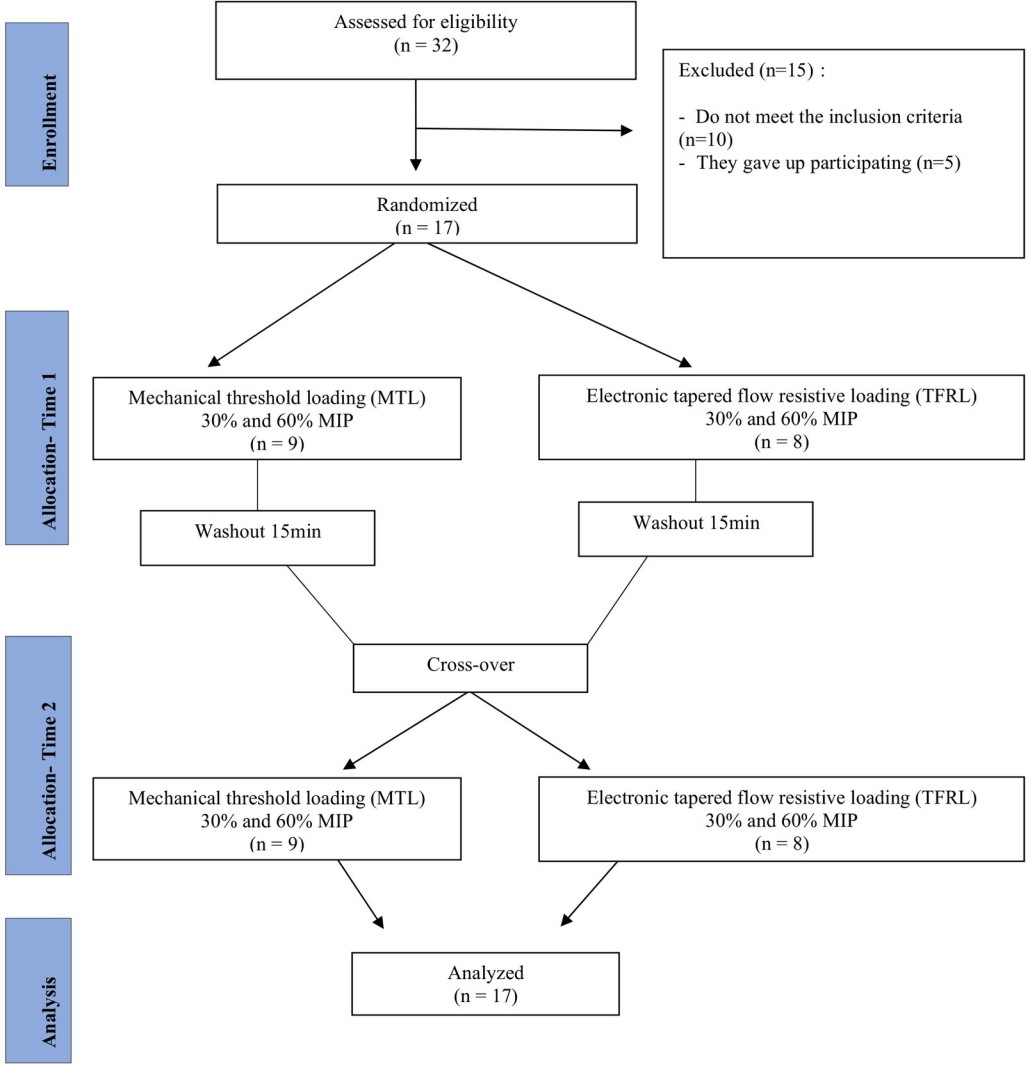

**Fig 1. Flow diagram of the study.** MIP, maximal inspiratory pressure.

(<80% of predicted forced vital capacity, or <70% of predicted forced expiratory volume in 1s) [19], history of exercise-induced asthma, and smokers were not included.

All evaluations and the study protocol were performed at the Physiology Laboratory of University of Brasília. Participants were evaluated using a standardized questionnaire covering personal and demographic characteristics, followed by a baseline clinical evaluation. The baseline assessment included: evaluation of muscle and fat mass by dual-energy X-ray absorptiometry (DXA) using a LUNAR Prodigy DXA-Scanner (GE Medical Systems, Version 6.70.021; Prodigy enCore2002, 726 Heartland Trail, USA); left ventricular eject fraction (LVEF) analysis [20] (M9, Mindray, CA, USA); and assessment of oxygen uptake ($VO_2$, mL/min) through the cardiopulmonary exercise test [21] (Quark CPET, Cosmed, Rome, Italy).

Next, for the assessment of pulmonary function, spirometry was carried out using a portable spirometer (Microlab 3.500, CareFusion, Yorba Linda, United States). According to ATS/ERS criteria [19], three forced expiratory maneuvers were performed in the sitting position, in a room with controlled temperature, ambient pressure, and relative humidity [22]. The variables analyzed were: Forced Vital Capacity (FVC, L), Forced Expiratory Volume in the first second ($FEV_1$, L) and (iii) the $FEV_1$/FVC ratio (%). The spirometry values obtained were recorded and compared to the predicted values for the Brazilian population [23].

Respiratory muscle strength was also measured. The MIP and maximal expiratory pressure (MEP) were obtained using a digital pressure transducer (MVD300[R], Globalmed. Porto Alegre, Brazil) previously calibrated with a measurement range of -300 to + 300 $cmH_2O$, with a smooth plastic mouthpiece connected to the device and the use of a nose clip to avoid air escape [7]. The S-Index, which is a dynamic assessment of inspiratory pressure [15,16,24], was measured using the POWERBreathe[R] KH2 device (International, Ltd., Warwickshire, United Kingdom). For the MIP and S-Index, both inspiratory muscle tests, participants were seated and instructed to perform a maximal voluntary exhalation at residual volume, followed by a maximal inspiratory effort, according to ATS/ERS standards [8]. The participants performed at least three maneuvers to assess the MIP, and ten maneuvers to assess the S-Index, measured with less than 10% deviation [7]. The highest value was used in the analysis. The MIP values obtained were recorded and compared with predicted values for the Brazilian population [25]. The MEP was performed with a maximal inhalation to total lung capacity (TLC) after which a verbal command was given to perform a maximal expiratory effort [7].

## Diaphragm thickness, diaphragm thickening fraction and mobility

On the second day, the diaphragm was assessed by a B-mode US device (M9, Mindray, CA, USA) to evaluate diaphragm thickness [4,10] and mobility [9,26,27], with the subject in a supine position with 30º of elevation [7].

For an accurate diaphragm thickness assessment, a 10 MHz linear transducer with high-resolution and low-penetration was placed on the anterior axillary line, to obtain a sagittal image of the intercostal space between the 7th to 9th ribs during inspiration and expiration using the intercostal access [4] In the zone of apposition, measurements of diaphragm thickness were obtained at the end of inspiration (Tdi) and at the end of expiration (Tde), during unload breathing and during loading, performing full-vital capacity maneuvers. The diaphragm thickening fraction (TFdi) was calculated, considering TFdi = (Tdi–Tde)/Tde [9,28] (Fig 2a).

To assess diaphragm mobility, a 3.5 MHz convex transducer was used, positioned over the right subcostal region. The craniocaudal displacement of the right diaphragm cupola, between residual volume and total lung capacity, was assessed at a perpendicular incidence angle to the craniocaudal axis [29–31]. Initially, the left branch of the portal vein was identified, then its position was marked with a cursor during forced expiration and forced inspiration, to identify

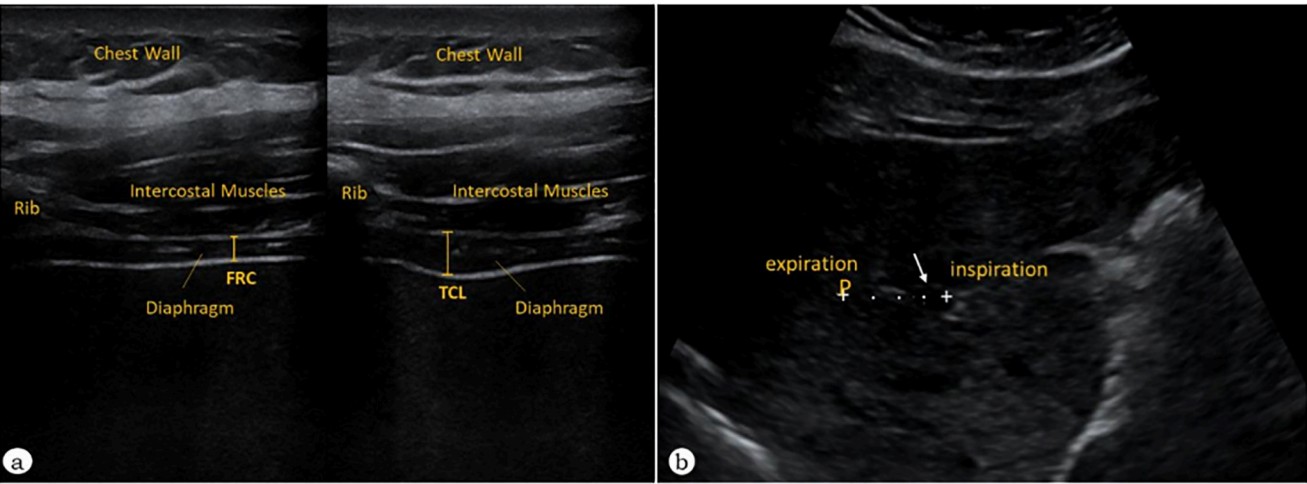

**Fig 2. Diaphragmatic assessment through Ultrasonography.** a. B-mode image; arrowheads mark the pleura and peritoneum (white lines) delimiting the right appositional area of a normal diaphragm at Functional Residual Capacity (FRC) on the left and at Total Lung Capacity (TLC) on the right. b. Indirect Technique- Ultrasound evaluation of craniocaudal displacement of the left branch of the hepatic portal vein. The vessel position was marked with the caliper during forced inspiration and expiration. Craniocaudal displacement of these points was recorded in millimeters and recorded as the degree of right diaphragmatic mobility.

the craniocaudal displacement of these points (in millimeters), which correspond to an indirect [4] and reproductive technique for measuring diaphragm mobility [9,32] (Fig 2b).

All evaluations were performed by the same evaluator, and the images were analyzed using the ImageJ software (SciJava) [33]. Ten measurements were recorded for each variable during each breathing condition and the highest value of 3 reproducible measurements (those with a variation lower at 5%) was used for the analysis [27].

## Study protocol

Using the sealed, serial, and opaque envelope method, the participants were randomized into two groups for the inspiratory effort protocol, which consisted of an acute loading test: group I, MTL (Plus, PowerBreathe®, London, England, UK) and group II, electronic TFRL (KH2, PowerBreathe®, London, England, UK). They were then placed in the supine position with a bedside elevation of 30˚. One researcher was responsible for the randomization and the application of the inspiratory effort protocol, with a different researcher assessing the diaphragm by US. A blinded researcher subsequently performed the data analyses.

For the assessment of the US baseline variables, participants performed 30 unloaded breaths at full vital capacity inspiration starting from residual volume, without any device, wearing a nose clip. After resting for 5 minutes, participants performed 30 breaths in the allocated modalities, with low-intensity (30% MIP) or high-intensity (60% MIP)—which was randomly assigned—concomitantly to the US diaphragm assessment. After a 15-minute washout rest period, participants changed modalities, and reproduced the same protocol (Fig 1). In addition, the Borg Rate of Perceived Exertion (RPE), according to the Borg CR10 scale, was assessed immediately after each inspiratory loading session.

## Statistical analysis

Results were summarized as means (*SD*). We tested the normality of the distribution through the Shapiro-Wilk test (S1 Table and S1 Fig). To analyze the difference in the diaphragm

thickness, diaphragm thickening fraction and mobility between loads, modalities, and rate of perceived exertion, paired Student t test with Bonferroni correction multiple tests were used. Associations were analyzed using Pearson's correlation (two-tailed), considering an absolute magnitude for the correlation coefficient of .00-.10 to be negligible; .10-.39 weak; .40-.69 moderate; .70-.89 strong and .90–1.00 very strong [34].

All data (S1 File) analyses were performed using a statistical software package (Graph Pad Prism, version 6.0, San Diego, CA). A $p$-value $\leq.05$ was considered statistically significant for all tests.

Considering the total sample size of 17 subjects, the analysis for the primary outcomes of the study revealed the power for the Tdi (mm) of 63%, Tfdi (%) of 76% and mobility (mm) of 97%.

## Results

From an initial group of 32 participants, recruited from January 5 to June 30, 2018; 15 were excluded according to the exclusion criteria: smoking ($n = 4$), infections disease ($n = 3$) and angina ($n = 3$), and being uninterested in participating in the study ($n = 5$) (Fig 1). Accordingly, 17 participants were included in the study.

The demographic data, initial cardiopulmonary measurements, and pulmonary function test results are presented in Table 1, as the baseline characteristics. The patients enrolled in the study did not undergo open-heart surgery or cardiac ablation procedures, which could potentially affect diaphragmatic function. Males represented the majority of the sample, the most predominant HF etiology was ischemic with mean left ventricular eject fraction of 50±13%, low exercise capacity, normal pulmonary function and inspiratory muscle test value higher than 80% of predicted MIP. All subjects had their pharmacological treatment optimized at least three months before the protocol. Fifteen were receiving Beta-blockers (88%); 13 ACE-inhibitors (76%); 6 angiotensin receptor blockers (35%); 6 digitals (35%) and 10 were receiving Loop diuretics (58%).

### Diaphragm thickness, diaphragm thickening fraction and mobility

Values of Tdi, measured during low and high-intensity loads using MTL and TFRL modalities, were significantly greater than values measured during unloaded breathing. Equally, values of TFdi were significantly greater during loaded breathing than those of unloaded breathing. Values of Tdi at high-intensity using TFRL were greater than low-intensity. However, values of TFdi at high-intensity, using both MTL e TFRL, were greater than low-intensity. All data is summarized in Table 2.

Diaphragm mobility measured during MTL and TFRL modalities of loaded breathing produced a significant increase under low-intensity loads ($\Delta$ = 14.85 [18.02, 11.67] and 16.26mm [19.43, 13.08], respectively) and high-intensity loads ($\Delta$ = 20.75 [23.92, 17.57] and 23.23mm [26.40, 20.05], respectively), compared to unloaded breathing (Table 2).

In addition, Rate of Perceived Exertion values were higher for the high-intensity load compared with the low-intensity load (4.12±1.5; 2.47±1.5; $p$ = .009) during loaded breathing with MTL. No differences were found for the same comparison during TRFL (4.35±2.17; 3.17±1.7, respectively). No differences were found for dyspnea comparing both modalities ($p>.05$).

Furthermore, during unloaded breathing, and assuming that correlation does not imply causation, Tdi was positively and moderately correlated with respiratory muscle tests: MIP, S-Index and MEP. A positive moderate correlation was also found between Tdi during unloaded breathing and FVC (Fig 3). Concerning TFdi, no statistically significant correlation was found with inspiratory muscle strength ($p>.05$) or pulmonary function ($p>.05$).

**Table 1. Demographic, baseline values of cardiopulmonary variables, pulmonary function tests and ultrasonographic data of heart failure subjects.**

| Variables | HF subjects |
|---|---|
| | ($n$ = 17) |
| Male sex (n, %) | 11 (65) |
| Age (years) | 55.1 (7.6) |
| BMI (Kg/m$^2$) | 27.6 (3.8) |
| Muscle Mass (%) | 48.1 (4.1) |
| Fat mass (%) | 35.1 (6.4) |
| HR (beats/min) | 72 (10) |
| SpO$_2$ (%) | 97 (1.1) |
| Ischemic/nonischemic (%) | 12/5 (70) |
| LVEF (%) | 50 (13) |
| NYHA (I-II/III-IV) | 11/6 |
| VO$_2$ peak (ml/kg$^{-1}$ · min) | 14.4 (4.6) |
| **Pulmonary Function Tests** | |
| FVE$_1$ (% predicted) | 76.1 (15.1) |
| FVC (L) | 3.22 (0.8) |
| FVC (% predicted) | 81.8 (12.9) |
| FEV$_1$/FVC | 0.89 (0.12) |
| MIP (cmH$_2$O) | 84.6 (35.7) |
| MIP (% predicted) | 83.0 (25.1) |
| MEP (cmH$_2$O) | 88.2 (34.2) |
| MEP (% predicted) | 82.1 (25.4) |
| S-Index (cmH$_2$O) | 86.6 (32.7) |
| **Ultrasonography data** | |
| Diaphragm thickness | |
| Tdi (mm) | 2.2 (0.3) |
| Tde (mm) | 1.9 (0.3) |
| TFdi (%) | 16.8 (7.3) |
| Diaphragm mobility (mm) | 19.3 (1.6) |

Data are expressed as mean (*SD*) or frequency (%). BMI, body mass index; HR, heart rate; SpO$_2$, pulse % oxygen saturation; LVEF, left ventricular eject fraction; NYHA, New York Heart Association; VO$_2$ peak, Peak oxygen uptake; MIP, maximal inspiratory pressure; S-Index, maximal dynamic inspiratory pressure; FVC, forced vital capacity; FEV$_1$, forced expiratory volume in 1s; FEV$_1$/FVC, ratio of FEV$_1$ to FVC; Tdi, Thickness of the diaphragm at the end of inspiration; Tde, Thickness of the diaphragm at the end of expiration; TFdi, Diaphragm thickening fraction.

## Discussion

The findings of the present study were: 1) low and high-intensity loads provided by both MTL and TFRL modalities induced greater diaphragm thickness and thickening fraction, higher at high-intensity comparing with low (except Tdi at MTL); 2) both TFRL and MTL modalities produced greater diaphragm mobility during low and high-intensity loading compared to baseline breathing, higher at high-intensity comparing with low; 3) diaphragm thickness showed moderate correlations with the respiratory muscle tests and FVC.

This appears to be the first study to analyze diaphragmatic thickness and mobility under different inspiratory loads and with different modalities of inspiratory muscle loading in a heart failure population.

**Table 2. Diaphragm thickness, diaphragm thickening fraction and mobility of heart failure subjects using different loads and devices.**

| Diaphragm | Baseline | MTL<br>n = 17 | | TFRL<br>n = 17 | |
|---|---|---|---|---|---|
| | | Low intensity 30% | High intensity 60% | Low intensity 30% | High intensity 60% |
| **Tdi (mm)** | 2.21 (0.26) | 2.53 (0.31)* | 2.68 (0.33)* | 2.47 (0.42)* | 2.73 (0.44)*† |
| **TFdi (%)** | 16.84 (7.30) | 30.19 (12.42)* | 41.57 (14.50)*† | 27.57 (12.63)* | 39.23 (12.75)*† |
| **Mobility(mm)** | 19.35 (1.66) | 34.20 (3.50)* | 40.10 (5.40)*† | 35.61 (2.93)* | 42.58 (2.29)*† |

Data are expressed as mean (*SD*). Tdi, Thickness of the diaphragm at the end of inspiration; TFdi, Diaphragm thickening fraction; MTL, Mechanical threshold loading; TRFL, Electronic tapered flow-resistive loading. Paired Student t test with Bonferroni correction for multiple tests (8 pairwise comparisons: 1. Baseline vs MTL-low; 2. Baseline vs MTL-high; 3. Baseline vs TFRL-low; 4. Baseline vs TFRL-high; 5. MTL-low vs MTL-high; 6. TFRL-low vs TFRL-high; 7. MTL-low vs TFRL-low; 8. MTL-high- vs TFRL-high).

*significant difference when compared to baseline.

†significant difference when compared to baseline to 30%.

Heart failure has been linked to inspiratory muscle dysfunction associated with oxygen consumption, functional status (NYHA), and age [35]. An impairment in diaphragm function represents inspiratory muscle weakness and is associated with reduced performance during exercise [6]. The findings showed that participants had a moderate degree of functional impairment yet did not present pulmonary function limitations nor lower inspiratory muscle strength [26,35].

Diaphragm thickness at end inspiration was different between unloaded breathing and loaded breathing with both low and high-intensity load. Increasing diaphragm thickness, during inspiratory, has been used as an indirect measurement of muscle fiber contraction [36]. It can also be used to monitor the evolution of diaphragm weakness, inspiratory muscles efficiency and respiratory workload [7,10]. Likewise, TFdi is considered a reliable parameter for diaphragm function evaluation [4] and it has been used to identify diaphragm paralysis [4,7]. The use of non-invasive ventilation with increasing levels of pressure support ventilation leads

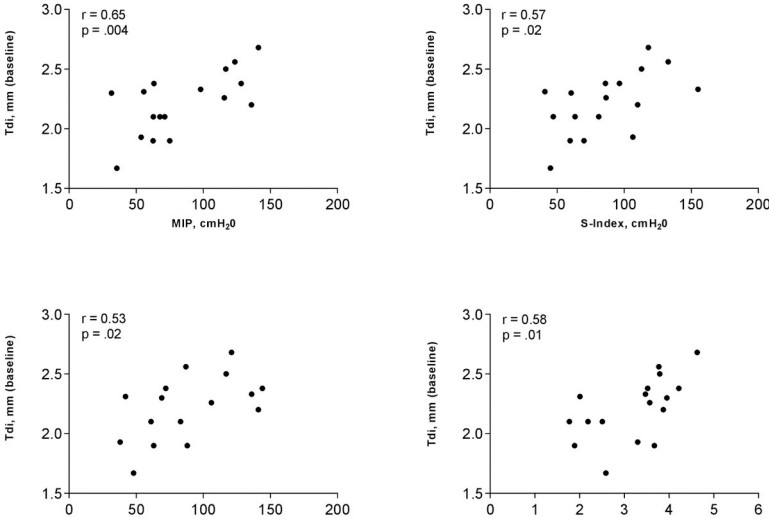

**Fig 3. Correlation between diaphragm thickness (Tdi) and respiratory variables.** MIP, maximal inspiratory pressure; S-Index, dynamic inspiratory pressure; MEP, maximal expiratory pressure; FVC, forced vital capacity. Data are presented in mean ± *SD*. Pearson's correlation.

to a decrease in the TFdi, suggesting its relationship with reduced respiratory effort [37]. In another study with HF patients, Tdi increased after IMT, with 60% of MIP, using the MTL device [10], similar to the present findings. Also, in HF patients, the reduction in the TFdi [6] suggested the coexistence of a diaphragmatic muscle myopathy related to exercise intolerance.

Regarding diaphragm mobility, greater mobility was associated with higher inspiratory loads in the participants, compared to low intensity. Diaphragm mobility has been considered another important outcome to evaluate the diaphragm function, according to US analysis [9], being related to inspiratory muscle function in chronic patients [9]. Also, greater diaphragm excursion was found in older adult women following an IMT training program with 40% MIP load [27]. No other study was found that evaluated diaphragm mobility with loads in HF patients, which hinders a comparative analysis.

Of the devices studied, the TFRL has dynamic-adjusted resistance provided by an electronic valve. The results showed that the IMT performed with TFRL allow similar displacement volume at equal intensity of loads compared to MTL, which is an important parameter to consider when evaluating the effects of IMT on respiratory muscle function [14].

The temporal increase of blood flow to the diaphragm muscle can also acutely increase the muscle thickness and facilitate the excursion of the diaphragm muscle. Its relationship with greater muscle activation has been also demonstrated [38].

Interestingly, the rated of perceived exertion (RPE) was similar between the devices using both low and high-intensity loads. Both devices require effort from the patient to overcome the load imposed by the system. The MTL depends on the flow and strength, while the TFRL depends on the flow and inspiratory volume [14].

Finally, a correlation was found between Tdi at unloaded breathing and respiratory muscle strength, as well as FVC. This finding strongly supports the use of US as a complimentary assessment to evaluate the mass, contractions, and function of the diaphragm in subjects with HF. In recent years, US has become a promising, simple, portable, non-invasive, ionization-free, real-time technique for disease detection and for monitoring the development of new diseases [39,40]. Moreover, it is correlated with pulmonary function and inspiratory strength, capable of detecting inspiratory muscle weakness, a factor associated with the development of exercise limitation in patients with heart failure [6].

Accordingly, further investigation of the clinical utility of US in HF and other subject populations with diaphragmatic dysfunction should be performed.

The clinical implications of IMT utilizing the MTL device, or an TFRL device, holds clinical significance by facilitating the optimization of therapeutic interventions aimed at improving pulmonary function, respiratory muscle strength, and enhanced performance during exercise. The TFRL and MTL devices has the capacity to induce greater diaphragm thickness and mobility under high-intensity training loads. As the results were similar, both devices contribute beneficially. The choice between them will depend on the specified needs of the individual patient.

## Limitations

The study presents several limitations: 1) the study had a small sample size and was conducted only with HF patients, which may limit the external validity of the study. However, these discoveries could be relevant, since HF subjects are susceptible to respiratory muscle weakness, and better knowledge about diaphragm function evaluation is needed; 2) the HF subjects in this study had no respiratory muscle weakness; however, we were able to evaluate the diaphragm function in the HF subjects without respiratory conditions, avoiding any overlap

influence; 3) we were unable to evaluate the tidal volume during inspiratory muscle loading with different load intensities, as the required adaptations to the device in order to perform the measures would mischaracterize the real daily life use of the device; 4) we could not develop the Motor Evoked potencies and Electromyography; as no equipment was available to perform these techniques; and 5) the measurement of diaphragm mobility was carried out using an indirect technique, which is somewhat limited by the intrinsic relationship with abdominal organs, however, was found to be similar to the other direct methods. Accordingly, further investigation is recommended to address these study limitations.

## Conclusion

The MTL and TFRL devices elicit similar increases in diaphragm thickness and thickening fraction during low and high-intensity loading. Both the TFRL and MTL modalities produced greater diaphragm mobility during low and high-intensity loading compared to baseline breathing. Diaphragm thickness showed moderate correlations with respiratory muscle tests and with FVC. Both devices are recommended.

## Supporting information

**S1 Fig. Normality test plot of variables.** Shapiro-Wilk test.
(DOCX)

**S1 Table. Normality test of variables.**
(DOCX)

**S1 File. Data set.**
(XLSX)

## Author Contributions

**Conceptualization:** Tatiana Zacarias Rondinel, Lilian Bocchi, Gerson Cipriano Júnior, Gaspar Rogério da Silva Chiappa, Gabriela de Sousa Martins, Sérgio Ricardo Menezes Mateus, Lawrence Patrick Cahalin, Graziella França Bernardelli Cipriano.

**Data curation:** Tatiana Zacarias Rondinel, Lilian Bocchi, Gabriela de Sousa Martins, Sérgio Ricardo Menezes Mateus.

**Formal analysis:** Tatiana Zacarias Rondinel, Lilian Bocchi, Gerson Cipriano Júnior, Gaspar Rogério da Silva Chiappa, Gabriela de Sousa Martins, Graziella França Bernardelli Cipriano.

**Investigation:** Tatiana Zacarias Rondinel, Lilian Bocchi, Gerson Cipriano Júnior, Gabriela de Sousa Martins, Graziella França Bernardelli Cipriano.

**Methodology:** Tatiana Zacarias Rondinel, Lilian Bocchi, Gerson Cipriano Júnior, Gaspar Rogério da Silva Chiappa, Gabriela de Sousa Martins, Sérgio Ricardo Menezes Mateus, Lawrence Patrick Cahalin, Graziella França Bernardelli Cipriano.

**Project administration:** Tatiana Zacarias Rondinel, Graziella França Bernardelli Cipriano.

**Resources:** Tatiana Zacarias Rondinel.

**Software:** Tatiana Zacarias Rondinel.

**Supervision:** Tatiana Zacarias Rondinel, Gerson Cipriano Júnior, Gaspar Rogério da Silva Chiappa, Sérgio Ricardo Menezes Mateus, Graziella França Bernardelli Cipriano.

**Validation:** Lilian Bocchi, Graziella França Bernardelli Cipriano.

**Visualization:** Tatiana Zacarias Rondinel, Gerson Cipriano Júnior, Lawrence Patrick Cahalin, Graziella França Bernardelli Cipriano.

**Writing – original draft:** Tatiana Zacarias Rondinel, Lilian Bocchi, Gaspar Rogério da Silva Chiappa, Gabriela de Sousa Martins, Sérgio Ricardo Menezes Mateus, Lawrence Patrick Cahalin, Graziella França Bernardelli Cipriano.

**Writing – review & editing:** Tatiana Zacarias Rondinel, Lilian Bocchi, Gerson Cipriano Júnior, Gaspar Rogério da Silva Chiappa, Gabriela de Sousa Martins, Lawrence Patrick Cahalin, Graziella França Bernardelli Cipriano.

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
