## [Decision Letter · Decision Letter 0]

3 Sep 2023

PONE-D-23-21191Diaphragm thickness and mobility elicited by two different modalities of inspiratory muscle loading in heart failure participants: a randomized crossover studyPLOS ONE

Dear Dr. Zacarias Rondinel,

Thank you for submitting your manuscript to PLOS ONE. After careful consideration, we feel that it has merit but does not fully meet PLOS ONE’s publication criteria as it currently stands. Therefore, we invite you to submit a revised version of the manuscript that addresses the points raised during the review process.

We look forward to receiving your revised manuscript.

Kind regards,

Yoshihiro Fukumoto

Academic Editor

PLOS ONE

Journal Requirements:

Reviewers' comments:

Reviewer's Responses to Questions

**Comments to the Author**

1. Is the manuscript technically sound, and do the data support the conclusions?

Reviewer #1: Partly

Reviewer #2: Yes

2. Has the statistical analysis been performed appropriately and rigorously? 

Reviewer #1: Yes

Reviewer #2: I Don't Know

3. Have the authors made all data underlying the findings in their manuscript fully available?

Reviewer #1: No

Reviewer #2: Yes

4. Is the manuscript presented in an intelligible fashion and written in standard English?

Reviewer #1: Yes

Reviewer #2: Yes

5. Review Comments to the Author

Reviewer #1: This study reports the relationships among diaphragmatic thickness, at end-inspiration and end-expiration, diaphragmatic thickening index and mobility via US under two different modalities of inspiratory muscle loading, in two different modalities of inspiratory muscle loading and different load intensities at full-vital capacity maneuvers and the relationship between diaphragmatic thickness with pulmonary function tests in participants with HF. Interesting study but I have several comments to the authors.

1. Could the authors provide a schema how they measure diaphragm thickness (Tdi, mm), fractional thickness (TFdi, %), and mobility by US that would be helpful to the readers who are not familiar with this topic.

2. Does the authors have data on the relationships among cardiac parameters (e.g. cardiac-echo parameters), physical activity (e.g. 6MWD and peak VO2) and Tdi, TFdi and mobility?

3. What is clinical implications? Could the authors provide clinical implications section in discussion?

Reviewer #2: The authors reported an ultrasonographic evaluation of the effects of two types of inspiratory muscle training on the diaphragm in HFpEF patients with normal pulmonary function.

Minor comments

1. Specify whether the patients had any prior history that would affect diaphragm function, e.g., after open heart surgery or cardiac ablation etc.

2. Page13, Line 286-288: Specify what functional impairment it is. This reviewer cannot determine from the context whether the functional impairment is heart failure or diaphragm.

3. This reviewer feel that the superiority of ultrasonographic assessment of the diaphragm in patients with heart failure compared to respiratory function tests and CPX should be added.

6. PLOS authors have the option to publish the peer review history of their article (what does this mean?). If published, this will include your full peer review and any attached files.

Reviewer #1: No

Reviewer #2: No

---

## [Author Response · Author response to Decision Letter 0]

23 Dec 2023

Reviewer #1:

C1: This study reports the relationships among diaphragmatic thickness, at end-inspiration and end-expiration, diaphragmatic thickening index and mobility via US under two different modalities of inspiratory muscle loading, in two different modalities of inspiratory muscle loading and different load intensities at full-vital capacity maneuvers and the relationship between diaphragmatic thickness with pulmonary function tests in participants with HF. Interesting study but I have several comments to the authors.

A1. We appreciate your effort in reviewing our manuscript, and we also thank you for making the below comments and suggestions.

C2. Could the authors provide a schema how they measure diaphragm thickness (Tdi, mm), fractional thickness (TFdi, %), and mobility by US that would be helpful to the readers who are not familiar with this topic.

A2: We appreciate your comment. We have incorporated additional information in "Fig. 2. Diaphragmatic Assessment through Ultrasonography," detailing the Ultrasonography assessment, including thickness (Tdi, mm), fractional thickness (TFdi, %), and mobility (Lines 159-166, page 07). 

Additionally, we have inserted a reference (Line 171, page 07).

Number 31: Yamaguti WP, Paulin E, Shibao S, Kodaira S, Chammas MC, Carvalho CR. Ultrasound evaluation of diaphragmatic mobility in different postures in healthy subjects. J Bras Pneumol. 2007 Jul-Aug;33(4):407-13. English, Portuguese. doi: 10.1590/s1806-37132007000400009.

C3. Does the authors have data on the relationships among cardiac parameters (e.g. cardiac-echo parameters), physical activity (e.g. 6MWD and peak VO2) and Tdi, TFdi and mobility?

A3. We thank you for its valuable comment; in fact, we had already analyzed these correlations. We did not find any differences when correlating LVEF (%), VO2 peak and VO2 max with TFdi, Tdi and Mobility. These correlations are shown below. 

Table. Correlation between cardiac US parameters and physical activity and TFdi (%), Tdi and Mobility. 

 TFdi (%) Tdi (mm) Mobility (mm)

 r p r p r p

LVEF (%) -0.17 0.495 0.18 0.484 -0.23 0.361

VO2peak 0.24 0.349 -0.04 0.851 -0.41 0.097

VO2máx 0.04 0.859 0.34 0.173 -0.23 0.356

TFdi, Fractional thickness of the diaphragm; Tdi, diaphragm thickness; LVEF, left ventricular eject fraction; VO2, oxygen uptake. Spearman’s correlation.

C4. What is clinical implications? Could the authors provide clinical implications section in discussion?

A4. Thank you for the suggestion for the clinical implications. As suggested, we addition of the information in discussion. Lines 348-355, pages 15 and 16.

Reviewer #2: 

C5: The authors reported an ultrasonographic evaluation of the effects of two types of inspiratory muscle training on the diaphragm in HFpEF patients with normal pulmonary function.

 A5. We appreciate your effort in reviewing our manuscript, and we also thank you for making the below comments and suggestions.

Minor comments

C6. Specify whether the patients had any prior history that would affect diaphragm function, e.g., after open heart surgery or cardiac ablation etc.

A6. Thank you for your question . The patients enrolled in the study did not undergo open-heart surgery or cardiac ablation procedures, which could potentially affect diaphragmatic function. Fifteen participants were excluded based on the exclusion criteria: smoking (n=4), infections disease (n=3) and angina (n=3) and being uninterested in participating in the study (n=5) (Fig.1).

C7. Page13, Line 286-288: Specify what functional impairment it is. This reviewer cannot determine from the context whether the functional impairment is heart failure or diaphragm.

A7. We appreciate your suggestion. We have revised the phrase to specify the functional impairment (at Line 300, page 14).

C8. This reviewer feel that the superiority of ultrasonographic assessment of the diaphragm in patients with heart failure compared to respiratory function tests and CPX should be added.

A8. We thank you for comment. We believe that diaphragmatic ultrasound emerges as a complementary technique for assessing the mass, contraction, and function of the diaphragm in individuals with heart failure. We have included additional information about US assessment in the text. (Lines 339- 345, page 15).

---

## [Decision Letter · Decision Letter 1]

22 Jan 2024

PONE-D-23-21191R1Diaphragm thickness and mobility elicited by two different modalities of inspiratory muscle loading in heart failure participants: a randomized crossover studyPLOS ONE

Dear Dr. Zacarias Rondinel,

Thank you for submitting your manuscript to PLOS ONE. After careful consideration, we feel that it has merit but does not fully meet PLOS ONE’s publication criteria as it currently stands. Therefore, we invite you to submit a revised version of the manuscript that addresses the points raised during the review process.

We look forward to receiving your revised manuscript.

Kind regards,

Yoshihiro Fukumoto

Academic Editor

PLOS ONE

Journal Requirements:

Reviewers' comments:

Reviewer's Responses to Questions

**Comments to the Author**

1. If the authors have adequately addressed your comments raised in a previous round of review and you feel that this manuscript is now acceptable for publication, you may indicate that here to bypass the “Comments to the Author” section, enter your conflict of interest statement in the “Confidential to Editor” section, and submit your "Accept" recommendation.

Reviewer #1: All comments have been addressed

Reviewer #2: All comments have been addressed

Reviewer #3: (No Response)

2. Is the manuscript technically sound, and do the data support the conclusions?

Reviewer #1: Yes

Reviewer #2: Yes

Reviewer #3: Partly

3. Has the statistical analysis been performed appropriately and rigorously? 

Reviewer #1: Yes

Reviewer #2: I Don't Know

Reviewer #3: No

4. Have the authors made all data underlying the findings in their manuscript fully available?

Reviewer #1: Yes

Reviewer #2: Yes

Reviewer #3: Yes

5. Is the manuscript presented in an intelligible fashion and written in standard English?

Reviewer #1: Yes

Reviewer #2: Yes

Reviewer #3: Yes

6. Review Comments to the Author

Reviewer #1: (No Response)

Reviewer #2: The authors' response is appropriate but not described in the manuscript. The authors should add the following text to the Results. The patients enrolled in the study did not undergo open-heart surgery or cardiac ablation procedures, which could potentially affect diaphragmatic function.

Reviewer #3: I have some comments on the statistical methods employed.

Bearing in mind that correlation does not imply causation (and I feel that this should be clearly stated in the manuscript), please provide evidence that all the assumptions underlying the employed parametric tests are met.

The authors state that they checked for the Gaussianity of the data distributions, but no evidence was provided.

From the figures showing the correlations across variables, the risk of interpreting it as causation is clear. Please, remove the straight lines from the graphs. Moreover, at first look, heteroscedasticity may arise.

Accordingly, as supplementary material, please show that:

- the data follow a Gaussian distribution;

- there is homoscedasticity;

– outliers are not present, they can significantly skew the correlation coefficient and make it inaccurate.

All these are fundamental to avoid misleading inference, as the sample size is rather tiny.

The ANOVA might not be the most suited method for the data at hand. Due to the clear difference between the baseline values and the others, it is not surprising that the p-value is less than .05 (given that the underlying assumptions are met). I strongly suggest to consider regression modelling, with univariable analyses first and with interactions then, to better appreciate between groups differences. Of course, the Gauss-Markov assumptions should be checked as well.

7. PLOS authors have the option to publish the peer review history of their article (what does this mean?). If published, this will include your full peer review and any attached files.

Reviewer #1: No

Reviewer #2: No

Reviewer #3: No

---

## [Author Response · Author response to Decision Letter 1]

22 Mar 2024

6. Review Comments to the Author

Reviewer #1: (No Response)

Reviewer #2: The authors' response is appropriate but not described in the manuscript. The authors should add the following text to the Results. The patients enrolled in the study did not undergo open-heart surgery or cardiac ablation procedures, which could potentially affect diaphragmatic function.

A: Thank you for the comment. As suggested, we added the sentence “The patients enrolled in the study did not undergo open-heart surgery or cardiac ablation procedures, which could potentially affect diaphragmatic function” at section Results, page 9, line 221-223.

Reviewer #3: I have some comments on the statistical methods employed.

Bearing in mind that correlation does not imply causation (and I feel that this should be clearly stated in the manuscript), please provide evidence that all the assumptions underlying the employed parametric tests are met.

The authors state that they checked for the Gaussianity of the data distributions, but no evidence was provided.

A: Thank you for the observation. We added, as suggested, the sentence “and assuming that correlation does not imply causation,” at section Results, page 13, line 276-277. Regarding data distributions of this data, we provided below a table with normality test (Shapiro-Wilk). Also, we elaborated a Supplementary Material with normality test represented as Figure and Table.

Shapiro-Wilk test Tdi Rest MIP FVC S-Index MEP

W 0,9789 0,9141 0,9174 0,9614 0,9434

P value 0,9464 0,1176 0,1336 0,6571 0,3608

Passed normality test (alpha=0.05)? Yes Yes Yes Yes Yes

P value summary ns ns ns ns ns

From the figures showing the correlations across variables, the risk of interpreting it as causation is clear. Please, remove the straight lines from the graphs. Moreover, at first look, heteroscedasticity may arise.

A: As suggested, we removed straight lines from correlations graphs in Figure 3. 

Accordingly, as supplementary material, please show that:

- the data follow a Gaussian distribution;

- there is homoscedasticity; 

– outliers are not present, they can significantly skew the correlation coefficient and make it inaccurate.

A: As suggested, we build a Supplementary Material, that shows the normal distribution of data and the absence of outliers, demonstrated at Tabel 1 e Figure 1. Since the analysis was revised considering the Paired Student t test for pairwise comparisons (see the question below), it is no longer necessary to check for equality of variances.

All these are fundamental to avoid misleading inference, as the sample size is rather tiny.

The ANOVA might not be the most suited method for the data at hand. Due to the clear difference between the baseline values and the others, it is not surprising that the p-value is less than .05 (given that the underlying assumptions are met). I strongly suggest to consider regression modelling, with univariable analyses first and with interactions then, to better appreciate between groups differences. Of course, the Gauss-Markov assumptions should be checked as well.

A: Thank you for your comment and, in fact, ANOVA is not the best test for this analysis, as the data are paired. In this context, the analysis was revised considering the Paired Student t test with Bonferroni correction for multiple tests (added at Statistical Analysis section, page 9, line 204-205). A total of 8 pairwise comparisons were performed:

1. Baseline vs MTL-low;

2. Baseline vs MTL-high;

3. Baseline vs TFRL-low;

4. Baseline vs TRFL-high;

5. MTL-low vs MTL-high;

6. TFRL-low vs TFRL-high;

7. MTL-low vs TFRL-low;

8. MTL-high- vs TFRL-high.

As a new statistical test was applied, we rewrite the pharagrafh at Diaphragm thickness, diaphragm thickening fraction and mobility section at Results, page 11, lines 243-248. Changes at Table 2 were added at Results section, line 257, page 12. Also, this information of statistical tests was added at Table’s legend, at line 260, page 12. Also, according to the analysis, we adjust the text at Discussion and Conclusion section. 

Furthermore, due to the paired observations, a regression analysis is also not suitable, since the same individual participates in both groups and to different intensities. Regarding the question of normality, the Shapiro-Wilk test did not reject the normality of the data.

---

## [Decision Letter · Decision Letter 2]

11 Apr 2024

Diaphragm thickness and mobility elicited by two different modalities of inspiratory muscle loading in heart failure participants: a randomized crossover study

PONE-D-23-21191R2

Dear Dr. Zacarias Rondinel,

We’re pleased to inform you that your manuscript has been judged scientifically suitable for publication and will be formally accepted for publication once it meets all outstanding technical requirements.

Kind regards,

Yoshihiro Fukumoto

Academic Editor

PLOS ONE

Additional Editor Comments (optional):

Reviewers' comments:

Reviewer's Responses to Questions

**Comments to the Author**

1. If the authors have adequately addressed your comments raised in a previous round of review and you feel that this manuscript is now acceptable for publication, you may indicate that here to bypass the “Comments to the Author” section, enter your conflict of interest statement in the “Confidential to Editor” section, and submit your "Accept" recommendation.

Reviewer #3: All comments have been addressed

2. Is the manuscript technically sound, and do the data support the conclusions?

Reviewer #3: (No Response)

3. Has the statistical analysis been performed appropriately and rigorously? 

Reviewer #3: (No Response)

4. Have the authors made all data underlying the findings in their manuscript fully available?

Reviewer #3: (No Response)

5. Is the manuscript presented in an intelligible fashion and written in standard English?

Reviewer #3: (No Response)

6. Review Comments to the Author

Reviewer #3: (No Response)

7. PLOS authors have the option to publish the peer review history of their article (what does this mean?). If published, this will include your full peer review and any attached files.

Reviewer #3: No

---

## [Editor Report · Acceptance letter]

7 May 2024

PONE-D-23-21191R2 

PLOS ONE

Dear Dr. Zacarias Rondinel, 

I'm pleased to inform you that your manuscript has been deemed suitable for publication in PLOS ONE. Congratulations! Your manuscript is now being handed over to our production team.

Kind regards, 

on behalf of

Dr. Yoshihiro Fukumoto 

Academic Editor

PLOS ONE